# Deficiency of ChPks and ChThr1 Inhibited DHN-Melanin Biosynthesis, Disrupted Cell Wall Integrity and Attenuated Pathogenicity in *Colletotrichum higginsianum*

**DOI:** 10.3390/ijms242115890

**Published:** 2023-11-02

**Authors:** Lingtao Duan, Li Wang, Weilun Chen, Zhenrui He, Erxun Zhou, Yiming Zhu

**Affiliations:** Guangdong Province Key Laboratory of Microbial Signals and Disease Control, College of Plant Protection, South China Agricultural University, Guangzhou 510642, China; duanlingtao2022@163.com (L.D.); wwwlll0802@163.com (L.W.); chenweilun0828@163.com (W.C.); zhenruihe@163.com (Z.H.)

**Keywords:** *Colletotrichum higginsianum*, DHN-melanin, polyketide synthase (Pks), trihydroxynaphthalene reductase (Thr1), pathogenicity

## Abstract

*Colletotrichum higginsianum* is a major pathogen causing anthracnose in Chinese flowering cabbage (*Brassica parachinensis*), posing a significant threat to the Chinese flowering cabbage industry. The conidia of *C. higginsianum* germinate and form melanized infection structures called appressoria, which enable penetration of the host plant’s epidermal cells. However, the molecular mechanism underlying melanin biosynthesis in *C. higginsianum* remains poorly understood. In this study, we identified two enzymes related to DHN-melanin biosynthesis in *C. higginsianum*: ChPks and ChThr1. Our results demonstrate that the expression levels of genes *ChPKS* and *ChTHR1* were significantly up-regulated during hyphal and appressorial melanization processes. Furthermore, knockout of the gene *ChPKS* resulted in a blocked DHN-melanin biosynthetic pathway in hyphae and appressoria, leading to increased sensitivity of the *Chpks*Δ mutant to cell-wall-interfering agents as well as decreased turgor pressure and pathogenicity. It should be noted that although the *Chthr1*Δ mutant still exhibited melanin accumulation in colonies and appressoria, its sensitivity to cell-wall-interfering agents and turgor pressure decreased compared to wild-type strains; however, complete loss of pathogenicity was not observed. In conclusion, our results indicate that DHN-melanin plays an essential role in both pathogenicity and cell wall integrity in *C. higginsianum*. Specifically, ChPks is crucial for DHN-melanin biosynthesis while deficiency of ChThr1 does not completely blocked melanin production.

## 1. Introduction

*Colletotrichum*, a member of the Ascomycete phylum, is a kind of widespread phytopathogen that can infect various crops worldwide [1]. In 2012, it was recognized as one of the top 10 fungal pathogens due to its scientific and economic significance [1]. Anthracnose caused by *Colletotrichum higginsianum* poses a significant threat to Chinese flowering cabbage (*Brassica parachinensis*) cultivation in South China, leading to reduced crop quality and yield [2]. The annual yield losses due to anthracnose outbreaks can reach up to 40%, severely impacting the development of the Chinese flowering cabbage industry in this region [2]. *C. higginsianum* has a hemibiotrophic lifestyle and has been found to infect various cruciferous plants including model plant *Arabidopsis* [3,4,5]. Moreover, genomic and transcriptomic data for *C. higginsianum* have been published and are readily accessible for genetic manipulation studies [6,7], making significant contributions to our understanding of the mechanisms underlying interactions between plant pathogenic fungi and host plants.

Melanin is a macromolecular pigment produced by various fungi through three pathways: polyketide (DHN), 3,4-dihydroxyphenyl-alanine (L-DOPA) and tyrosine degradation synthetic pathways [8,9,10]. Previous studies have demonstrated that *C. higginsianum* produces DHN-melanin through the polyketide synthase pathway [5]. The biosynthetic pathway involves the polymerization of acetyl-CoA and malonyl-CoA by polyketide synthase (Pks) to form 1,3,6,8-Tetrahydroxynaphthalene (T4HN). T4HN then undergoes enzymatic conversions catalyzed by tetrahydroxynaphthalene reductase (Thnr), scytalone dehydratase (Scd), and trihydroxynaphthalene reductase (Thr1) to produce vermelone. Finally, vermelone is dehydrated and polymerized to form DHN-melanin [10,11]. It has been observed in related fungal species that deficiencies or loss of enzymes involved in this pathway can result in blocked biosynthesis of DHN-melanin [8,12,13]. However, the specific role played by these enzymes in DHN-melanin biosynthesis in *C. higginsianum* remains poorly understood and requires further research.

Melanin is an integral component of the cell wall rather than an independent entity [14], capable of covalently cross-linking with polysaccharide components within it [15,16]. In fungi, there exists a direct interaction between melanin and chitin—a crucial constituent of the fungal cell wall [17]. Reports suggest that chitin acts as a scaffold for melanin deposition within the fungal cell wall contributing to its fixation [18,19]. Additionally, melanin deposited on the fungal cell wall effectively enhances its integrity against various environmental stresses such as heavy metals, ultraviolet (UV) irradiation, and enzymatic lysis [8,9,20]. For example, mutants lacking melanin in *Botrytis cinerea* exhibit increased sensitivity to agents that interfere with cell wall integrity [21]. In terms of pathogenicity, proper appressorium formation and maintenance of high turgor pressure are crucial for successful infection by fungi like *C. higginsianum* and *Magnaporthe oryzae* [22,23,24]. Melanization primarily depends on melanin deposition in the cell wall [25]. Impaired melanin synthesis in *M. oryzae* inhibits proper appressorium melanization, resulting in reduced intracellular turgor pressure and decreased pathogenicity due to failure in penetrating host plant epidermal cells [24]. Similarly, deficiency of the gene *THR1* in *C. lagenarium* impairs DHN-melanin production and nonmelanized appressoria formation leading to a notable reduction in pathogenicity [26]. Conversely, in *C. gloeosporioides*, deletion of *CgCmr1* and *CgPks1* lead to unmelanization and decreased turgor pressure in the appressoria, while had no effect in pathogenicity [27]. Given the unclear relationship between melanin, cell wall integrity, and pathogenicity in *C. higginsianum*, further research is crucial for elucidating the role of melanin in this particular species.

In this study, we aimed to investigate the role of two enzymes (ChPks and ChThr1) involved in DHN-melanin biosynthesis in *C. higginsianum*. The expression patterns of genes *ChPKS* and *ChTHR1* were analyzed using the qPCR technique. Gene knockout and complementation strains were constructed for both genes to assess their impact on DHN-melanin biosynthesis as well as growth, development, and pathogenicity in *C. higginsianum*. The main goal of this research is to gain a better understanding of the molecular mechanisms underlying DHN-melanin biosynthesis and its relationship with cell wall integrity and pathogenicity in *C. higginsianum*. This knowledge will contribute to improve our understanding of the pathogenesis in anthracnose disease of cruciferous plants. Ultimately, it can aid in developing more effective control strategies against this disease.

## 2. Results

### 2.1. Identification and Characterization of ChPks and ChThr1 in C. higginsianum

Using the BLASTp search on NCBI, we identified a homologue of the *C. graminicola* PKS1 protein (XP_008093079) in *C. higginsianum*, which we named ChPks (XP_018155915). ChPks consists of a coding sequence of 2198 amino acids in length and contains six characteristic domains, including ACP transacylase domain (SAT), acyl transferase domain (PksD), iterative type I PKS product template domain (PT_fungal_PKS), thioesterase domain (EntF) and two phosphopantetheine attachment site (PP) (Figure 1A). The results of phylogenetic tree analysis demonstrated that ChPks and Pks homologues from *C. graminicola*, *C. orbiculare* (TDZ26149) and *Verticillium dahliae* (XP_009649862) shared high similarity (Figure 1C).

Similarly, we performed a BLASTp search on NCBI using the *Sclerotinia sclerotiorum* Thr1 (XP_001586798) protein sequence and identified a homologue in *C. higginsianum*, referred to here as ChThr1 (XP_018155910). ChThr1 was 278 amino acids in length and had a rossmann-fold NAD(P)-binding domain (Figure 1B). Because trihydroxynaphthalene reductase (Thr1) and tetrahydroxynaphthalene reductase (Thnr) had high sequence similarity, we selected Thr1 and Thnr from a number of fungi for phylogenetic analysis. The results showed that ChThr1 aggregated with the Thr1 from other fungal species, and ChThr1 had high degree of similarity with Thr1 homologues from *C. fioriniae* (XP_007595358)*, V. alfalfae* (XP_003008893) and *V. dahliae* (XP_009649855) (Figure 1D).

### 2.2. Expression of Genes ChPKS and ChTHR1 Is Up-Regulated during Melanization

The hyphae of *C. higginsianum* gradually melanization after 7~10 d of incubation in PDB liquid medium. To assess the expression patterns of genes *ChPKS* and *ChTHR1* during hyphae melanization, we used real-time quantitative PCR (RT-qPCR) to determine the expression levels of genes *ChPKS* and *ChTHR1* in hyphae cultured for 7~10 d. The results showed a significant up-regulation in the expression levels of both genes, *ChPKS* and *ChTHR1*, during hyphal melanization (Figure 2A,C). Specifically, compared to 7 d, the expression level of gene *ChPKS* at 10 d was up-regulated by approximately 11 fold (Figure 2A), while gene *ChTHR1* was up-regulated by approximately 5 fold (Figure 2C).

Subsequently, the expression patterns of genes *ChPKS* and *ChTHR1* were investigated during infection. The results reveled a significant up-regulation in the expression levels of both genes after 8 h post-incubation (hpi) (Figure 2B,D), which coincided with the initiation of appressorium melanization. Furthermore, gene *ChTHR1* exhibited a significant up-regulation at 40 hpi (Figure 2D). Meanwhile, the ChThr1-mCherry signal was detected in appressoria during appressorium melanization and biotrophic infection phase (Appendix A). These findings suggest its potential involvement in the biotrophic infection phase beyond appressorium melanization of *C. higginsianum*.

Additionally, we determine the expression of genes *ChPKS* and *ChTHR1* using *β-TUBULIN* as the endogenous reference gene, which eventually resulted in an expression pattern similar to that with *ACTIN* as the endogenous reference gene (Appendix A).

### 2.3. ChPks and ChThr1 Are Involved in DHN-Melanin Biosynthesis

To evaluate the role of ChPks and ChThr1 in DHN-melanin biosynthesis in *C. higginsianum*, we employed ATMT (*Agrobacterium tumefaciens* mediated transformation) to knockout the genes *ChPKS* and *ChTHR1* in the WT strain (Appendix A). PCR was used to confirm that the gene *ChPKS* and *ChTHR1* were knockout in their respective mutants (Appendix A–F). Using the PKS-probe and the THR1-probe, a single hygromycin phosphotransferase gene was detected by Southern blotting to replace the genes *ChPKS* and *ChTHR1* in the *Chpks*Δ and *Chthr1*Δ mutant genomes, respectively (Appendix A). As depicted in Figure 3A, the colony of the *Chpks*Δ mutant exhibited a complete absence of melanization and displayed an orange-yellow coloration compared to the WT strain. Conversely, while the colony of the *Chthr1*Δ mutant displayed reduced melanization ability, it did not completely lose its capacity for DHN-melanin biosynthesis. The deficiencies observed in both mutants were restored in the complementation strains (Figure 3A). Additionally, TEM (transmission electron microscopy) analysis revealed that DHN-melanin deposits formed a dense layer on the cell wall of WT strains (Figure 4). In contrast, no melanin layer was detected on the cell wall of the *Chpks*Δ mutant (Figure 4), although there were still some reduced numbers of melanin deposits present on the cell wall of the *Chthr1*Δ mutant when compared to WT strains (Figure 4). Since both ChThr1 and ChThnr contain the conserved core motifs of the short-chain dehydrogenase/reductase (SDR) family, we speculate that ChThnr will partially supplement the role of ChThr1 in the *Chthr1*Δ mutant. However, the expression level of gene *ChTHNR* was notably down-regulated in the *Chthr1*Δ mutant (Appendix A).

Furthermore, we investigated the impact on vegetative growth caused by deficiencies in ChPks and ChThr1. After cultivation on PDA medium for 5 d, there was no significant difference in colony diameter between both the *Chpks*Δ and *Chthr1*Δ mutants compared to the WT strain (Figure 3B). These findings suggest that deficiency or disruption of either gene inhibits DHN-melanin synthesis; however, DHN-melanin synthesis is completely blocked only in the *Chpks*Δ mutant.

### 2.4. Melanin Is Essential for Cell Wall Integrity in C. higginsianum

Previous studies have shown that melanin accumulation enhances cell wall integrity, thereby helping the fungus to defend a wide range of abiotic stresses [8,9,10]. Therefore, we analyzed the growth of the WT, *Chpks*Δ, *Chthr1*Δ, *Chpks*Δ-C and *Chthr1*Δ-C strains under the stresses of CFW (calcofluor white), sodium dodecyl sulfate (SDS) and CR (congo red). After culturing for 5 d, we observed significantly increased growth inhibition rates compared to WT, *Chpks*Δ-C and *Chthr1*Δ-C strains (Figure 5A,B). These results indicate that the C*hpks*Δ and *Chthr1*Δ mutants displayed higher sensitivity to these cell-wall-interfering agents (CFW, SDS, CR). This suggests that the disruption of DHN-melanin biosynthesis caused by the knockout of genes *ChPKS* and *ChTHR1* affected the cell wall integrity of *C. higginsianum*.

### 2.5. Analysis of Conidiation, Appressorium Formation and Morphology, Turgor Pressure in ChpksΔ and Chthr1Δ Mutants

To further investigate the function of ChPks and ChThr1 in C. *higginsianum*, we conducted an analysis on the conidiation and appressorium formation rate of the *Chpks*Δ and *Chthr1*Δ mutant. The results revealed a significant decrease in conidiation for both the *Chpks*Δ and *Chthr1*Δ mutants compared to the WT strain. Specifically, when cultured on mathur medium for 5 d, the WT strain produced 8.92 × 10^6^·mL^−1^ conidia, while the conidiation of the *Chpks*Δ and *Chthr1*Δ mutants was only 4.83 × 10^6^·mL^−1^ and 4.38 × 10^6^·mL^−1^, respectively (Figure 6A). However, it is worth noting that neither absence of either ChPks nor absence of ChThr1 had any effect on appressorium formation rate (Figure 6B). We observed that only the appressoria of the *Chpks*Δ mutant lost their melanization ability compared to those formed by WT and *Chthr1*Δ strains (Figure 6C).

Due to the crucial role of DHN-melanin in appressorium turgor pressure accumulation, we conducted a glycerol cytorrhysis assay to assess the appressorium turgor pressure in the *Chpks*Δ and *Chthr1*Δ mutants. The results revealed that under 1 M glycerol concentration, the collapse rates of *Chpks*Δ and *Chthr1*Δ mutant appressoria were 33.1% and 13.6%, respectively, whereas WT, *Chpks*Δ-C and *Chthr1*Δ-C strains exhibited only approximately 3% collapse (Figure 6D). Similarly, at higher concentrations of glycerol (2 M and 3 M), both *Chpks*Δ and *Chthr1*Δ mutants displayed significantly higher rates of appressorium collapse compared to WT, *Chpks*Δ-C and *Chthr1*Δ-C strains. Additionally, it was observed that under conditions with a glycerol concentration of 1 M or 2 M, the appressoria collapse rate of *Chthr1*Δ mutant was notably higher compare to the WT strain (Figure 6D). In conclusion, our findings demonstrate that both ChPks and ChThr1 play essential roles in conidiation within *C. higginsianum*. The absence of DHN-melanin biosynthesis in the *Chpks*Δ mutant impedes appressorium melanization as well as turgor pressure accumulation. On the other hand, the *Chthr1*Δ mutant capable of forming melanized appressoria; however, it exhibits reduced turgor pressure potentially due to limited inhibition of DHN-melanin biosynthesis.

### 2.6. ChPks and ChThr1 Are Enssential for Pathogenicity in C. higginsianum

Previous studies have demonstrated that turgor pressure is important factor in the pathogenicity of *Colletotrichum* [23]. Consistent with these findings, our results revealed that a significant reduction in pathogenicity of both the *Chpks*Δ and *Chthr1*Δ mutants on *Arabidopsis* plants compared to the WT strain. Specifically, while leaves inoculated with the *Chpks*Δ mutant exhibited a complete loss of pathogenicity, those inoculated with the *Chthr1*Δ mutant still displayed some typical water-soaked, collapsed anthracnose lesions (Figure 7A). Furthermore, we observed that the *Chpks*Δ mutant also lost its ability to cause disease on detached leaves of Chinese flowering cabbage; however, when inoculated on wounded leaves, the pathogenicity was restored (Figure 7B). In contrast, there was no significant difference in pathogenicity between the *Chthr1*Δ mutant and the WT strain when tested on detached leaves of Chinese flowering cabbage (Figure 7B). These findings unequivocally demonstrate that both ChPks and ChThr1 play indispensable roles in mediating pathogenicity in *C. higginsianum*. Notably, while reduced turgor pressure severely impairs penetration ability in the case of the *Chpks*Δ mutant, it appears to have a less pronounced effect on this aspect for the *Chthr1*Δ mutant.

## 3. Discussion

Melanin is a critical component of fungal cell wall that helps to maintain cell wall integrity and is involved in against environmental stresses, accumulation of turgor pressure and pathogenicity [9,16]. Previous studies have reported that the genus *Colletotrichum* synthesizes DHN-melanin through the polyketide synthase pathway [5,27,28]. Deficiencies in these enzymes can inhibit or block DHN-melanin biosynthesis in many fungi [25,26,27]. Pks and Thr1 are key enzymes that play a crucial role in the biosynthetic processes of DHN-melanin in several fungi [29,30]. For example, in *C. graminicola, CgPKS1* gene encoding polyketide synthase is expressed during appressorial melanization [25]. Similarly, the expression level of *BRN2*, a homolog of THR1 in *Monilinia fructicola* and *M. fructigena,* is noticeably up-regulated during melanin synthesis [31]. In our study, we observed a significant up-regulation of both genes *ChPKS* and *ChTHR1* during hyphal and appressorial melanization processes of *C. higginsianum*. This finding indicates that ChPks and ChThr1 are essential players in DHN-melanin production. Interestingly, we also found that the expression level of gene *ChTHR1* was significantly up-regulated at 40 h post-infection (hpi), suggesting its potential involvement beyond appressorial melanization processes during the biotrophic infection phase of *C. higginsianum*.

Polyketide synthase (Pks) has been defined as the initial step in the DHN-melanin biosynthetic process [32,33,34]. Deletion of the *PKS1* gene in *C. graminicola* and *C. gloeosporioides* resulted in colonies that were yellow to light orange in color, indicating a lack of melanization. In addition, appressoria lacking *CgPKS1* in *C. graminicola* showed sensitivity to externally applied cell-wall-degrading enzymes [25,27]. Similarly, deletion of the *ChPKS* gene led to an orange-yellow colony and unmelanized appressoria. Furthermore, TEM observation revealed that the *Chpks*Δ mutant did not exhibit any deposits of DHN-melanin dots on its cell wall. This disruption affected the integrity of the cell wall and made it sensitive to agents interfering with cell wall function. Melanin deposition is known to be essential for maintaining turgor pressure within appressoria. For instance, *CgPks1*Δ mutants exhibited significantly lower turgor pressure compared to WT strains of *C. gloeosporioides*; however, *CgPks1* was found not to play an essential role in penetration and pathogenicity [27]. In contrast, knockout mutants lacking *CgPKS1* did not affect turgor pressure but noticeably reduced penetrance and pathogenicity in *C. graminicola* [25]. These findings indicate that Pks plays different roles among fungal species. In our study with *Chpks*Δ mutants from *C. higginsianum*, we observed a significant decrease in turgor pressure within appressoria similar to what was seen with deletion of *CgPks1* in *C. gloeosporioides* due to disrupted cell wall integrity resulting from unmelanized appressoria formation by *Chpks*Δ mutants. This loss of turgor pressure accumulation led to a loss of penetrance and pathogenicity.

Trihydroxynaphthalene reductase (Thr1) is also a key enzyme in DHN-melanin biosynthesis [35,36,37]. In our study, we observed that the knockout of the gene *ChTHR1* resulted in colony and appressoria with limited melanization, as well as a significant reduction in turgor pressure in the *Chthr1*Δ mutant. This phenotype is similar to that observed in *M. oryzae* mutants lacking the *Buf1* gene encoding trihydroxynaphthalene reductase; however, it should be noted that these Δ*buf1* mutants exhibit reduced penetrance [24]. Similarly, disruption of gene *THR1* also leads to loss of penetrance in *C. lagenarium* [26]. Interestingly, our results revealed that the *Chthr1*Δ mutant was still capable of penetrating Chinese flowering cabbage and *Arabidopsis* leaves. TEM observation revealed the cell wall of *Chthr1*Δ mutant fails to form a dense melanin layer, resulting in a notably reduction in turgor pressure and increased sensitivity to cell-wall-interfering agents compared to the WT strains. However, some melanin dots were still deposited on the cell wall of the *Chthr1*Δ mutant, leading to turgor pressure compared to *Chpks*Δ mutant. Notably though, this decreased turgor pressure did not affect penetrance. Furthermore, we observed a significant reduction in pathogenicity of *Chthr1*Δ mutants on *Arabidopsis* plants when compared to the WT strains. This finding suggests that while ChThr1 may play a role during biotrophic infection phase and contribute to pathogenesis in *C. higginsianum*, its exact mechanisms require further investigation.

During DHN-melanin biosynthesis of *C. higginsianum*, the absence of both ChPks and ChThr1 inhibit melanin production. However, compare to the *Chpks*Δ mutant where DHN-melanin biosynthetic processes is completely blocked, the *Chthr1*Δ mutant is still able to produce melanin. These results demonstrate that enzymes upstream in the DHN-melanin biosynthesis pathway may have a greater role in DHN-melanin synthesis. Additionally, it has been reported that both reductases Thnr and Thr1 belong to the short-chain dehydrogenase/reductase (SDR) family and share common conserved core structural motifs during DHN-melanin biosynthesis [29]. Studies conducted on *C. lagenarium* have revealed that Thnr and Thr1 can co-mediate the reduction of T4HN, suggesting functional complementarity between them [38]. However, our results indicate that expression levels of gene *ChTHNR* was down-regulated in the *Chthr1*Δ mutant (Appendix A), suggesting that additional mechanisms may be present in *C. higginsianum* to partially replace the role of ChThr1.

In summary, the deficiency of ChPks completely blocks DHN-melanin biosynthesis, thereby compromising the integrity and rigidity of the cell wall. Consequently, this results in a significant decrease in sensitivity to cell-wall-interfering agents as well as turgor pressure while also reducing pathogenicity in *C. higginsianum*. It is important to note that despite these effects on melanin production being observed in the *Chthr1*Δ mutant with limited DHN-melanin biosynthesis occurring within its hyphae and appressoria region, complete loss of pathogenicity has not been observed thus far.

## 4. Materials and Methods

### 4.1. Strains, Plant and Culture Conditions

The *C. higginsianum* IMI 349063 strain used in this study for whole genome sequencing was generously provided by Prof. Junbin Huang of Huazhong Agricultural University (Wuhan, China). All fungal strains were cultured at a temperature of 27 °C.

To assess colony morphology after 5 days cultivated in the dark, the wild type (WT) and all fungal strains were cultivated on PDA medium containing potato extract (200 g/L), dextrose (20 g/L), and agar (20 g/L).

For pathogenicity tests on *Arabidopsis thaliana* Col-0, the plants were grown for 4 weeks under controlled conditions with a temperature regime of 12 h light/12 h dark at temperatures of 22 °C during the day and 20 °C during the night.

Similarly, Chinese flowering cabbage was used for pathogenicity tests after incubating for 4 weeks under controlled conditions with a constant temperature of 26 °C with a photoperiod of 12 h light/12 h dark.

### 4.2. Bioinformatics Analysis of ChPks and ChThr1

ChPks protein sequence (XP_018155915) and ChThr1 protein sequence (XP_018155910) were retrieved from the genome database of *C. higginsianum* IMI 349063 using BLASTp search with *C. graminicola* PKS1 protein sequence (XP_008093079) and *Sclerotinia sclerotiorum* Thr1 protein sequence (XP_001586798) as queries. The conserved domains within ChPks and ChThr1 proteins were predicted using Pfam analysis available at http://pfam.xfam.org/ (accessed on 25 August 2023). A neighbor-joining phylogenetic tree was constructed based on amino acid alignments using MEGA X64 10.1.8 software.

### 4.3. Expression Pattern Analysis of Genes ChPKS and ChTHR1

The WT strain of *C. higginsianum* was inoculated in PDB (potato 200 g/L, dextrose 20 g/L) medium, and the hyphae was collected at 7, 8, 9 and 10 d for RNA extraction. The expression levels of genes *ChPKS* and *ChTHR1* were determined using real-time quantitative PCR (RT-qPCR), with *ACTIN* (CH63R_04240) and *β-TUBULIN* (CH63R_12878) serving as endogenous reference gene. The reverse transcription was performed using TransScript^®^ One-Step gDNA Removal and cDNA Synthesis SuperMix (TransGen Biotech, Beijing, China) according to protocol. The expression levels of genes *ChPKS* and *ChTHR1* were analyzed by RT-qPCR using the Bio-Rad CFX96 Real-Time PCR Detection System and ChamQ Universal SYBR qPCR Master Mix (Vazyme, Nanjing, China). The volume of the RT-qPCR reaction was 20 µL, including 10 µL of 2 × ChamQ Universal SYBR qPCR Master Mix, 0.4 µL Primer-F (10 µM), 0.4 µL Primer-R (10 µM), 2 µL of cDNA (100 ng/µL), and 7.2 µL of ddH_2_O. The RT-qPCR program consisted of an initial denaturation at 95 °C for 30 s, followed by 40 cycles at 95 °C for 10 s and at 60 °C for 30 s.

Additionally, conidia from the WT strain of *C. higginsianum* were collected and adjusted to a concentration of 1 × 10^6^ mL^−1^. Four-week-old *A. thaliana* Col-0 seedlings were used for inoculation. Samples were taken at various time points including 0 h post-inoculation (hpi), 8 hpi, 22 hpi, 40 hpi and 60 hpi for RNA extraction. RT-qPCR was used to analyze the expression pattern of genes *ChPKS* and *ChTHR1* during infection. Relative expression levels were calculated using the 2^−ΔΔCt^ method. The primers used are shown in Table 1.

### 4.4. Deletion and Complementation of ChPKS and ChTHR1

To knockout genes *ChPKS* and *ChTHR1*, we constructed knockout plasmids p821-ChPKS-KO and p821-ChTHR1-KO according to the methods described in Appendix A. Fragments of 1000 bp upstream and downstream of genes *ChPKS* and *ChTHR1* were amplified using primer pairs PKS-UP/DS and THR1-UP/DS, respectively. These fragments were then transformed into the ATMT vector pFGL821, resulting in the creation of knockout vectors for the genes *ChPKS* and *ChTHR1*, named p821-ChPKS-KO and p821-ChTHR1-KO. Subsequently, the knockout plasmid was transformed into the WT strain using the ATMT technique to replace genes *ChPKS* and *ChTHR1* with *HPH1* (hygromycin phosphotransferase gene), respectively. Hygromycin-resistant transformants were isolated as candidate mutants for *Chpks*Δ and *Chthr1*Δ. PCR analysis was used to confirm the *Chpks*Δ and *Chthr1*Δ mutants (Appendix A). The *Chpks*Δ and *Chthr1*Δ mutants were further confirmed by Southern blotting. Using the endonucleases *Sma* I and *Eco*R I to digest gDNA of WT and *Chpks*Δ mutants, and gDNA of WT and *Chthr1*Δ mutants were digested with *Acc*65 I. Then, agarose gel electrophoresis to separate the DNA fragment, and transferred to a nylon membrane and assayed using DNA probes specific for *ChPKS* and *ChTHR1*, respectively.

Subsequently, for complementation of genes *ChPKS* and *ChTHR1* at their original position, we constructed complementation plasmids pSFZY-ChPKS-COM and pSFZY-ChTHR1-COM according to the methods described in Appendix A. Fragments of 1001–2000 bp upstream, 1000 bp upstream + full length of gene *ChPKS* and 1000 bp downstream of gene *ChPKS* were amplified using primer pairs PKSc-F1, F2 and F3. These fragments were transformed into the ATMT vector pSFZY to create a complementary vector for the gene *ChPKS*, named pSFZY-ChPKS-COM. Additionally, fragments of 1001–2000 bp upstream, 1000 bp upstream + full length of gene *ChTHR1,* mCherry and 1000 bp downstream of gene *ChTHR1* were amplified using primer pairs PKSc-F1, F2, F3 and F4. Subsequently, these fragments were transformed into the ATMT vector pSFZY to create a complementary vector for the gene *ChTHR1*, named pSFZY-ChTHR1-COM. Similarly, we used the ATMT technique to transform pSFZY-ChPKS-COM and pSFZY-ChTHR1-COM into the *Chpks*Δ and *Chthr1*Δ mutants, respectively. Neomycin-resistant transformants were isolated as candidate complementation strains for *Chpks*Δ-C and *Chthr1*Δ-C. The presence of these complemented strains was confirmed through PCR analysis (Appendix A–F). The primers used are shown in Table 1.

### 4.5. Phenotypic Analysis

All fungal strains were cultured on 6 cm PDA plates at 27 °C under dark conditions, and the colony diameter was measured after 5 d of growth. Additionally, all fungal strains were grown on PDA plates supplemented with specific compounds: 200 μg·mL^−1^ calcofluor white (CFW), 0.01% sodium dodecyl sulfate (SDS) and 200 μg·mL^−1^ congo red (CR). The inhibition rate was calculated by measuring the colony diameter after 5 d of growth.

To determine conidiation, the concentration of conidia suspensions for all strains was adjusted to a density of 1 × 10^6^·mL^−1^, and spread onto 9 cm Mathur plates containing glucose (2.8 g·L^−1^), MgSO_4_·7H_2_O (1.22 g·L^−1^), KH_2_PO_4_ (2.72 g·L^−1^), oxoid mycological peptone (2.18 g·L^−1^), and agar (30 g·L^−1^). After incubation in darkness at 27 °C for 5 days, conidia were washed down with 5 mL of sterile distilled water. Conidiation was determined by measuring the concentration of conidial suspensions using a hemocytometer.

For appressorial formation analysis, the concentration of conidia suspensions for all strains was adjusted to 1 × 10^5^·mL^−1^. Then, a volume of 20 μL conidia suspensions from each strain was placed on hydrophobic cover glass. After incubating in darkness at 27 °C for 22 h, the rate of appressorial formation was observed and recorded under the microscope. The formation rate of at least 100 appressoria per strain should be assessed.

To assess appressorial turgor, conidia suspensions from all strains were collected and adjusted to a concentration of 1 × 10^5^·mL^−1^. Subsequently, 20 μL of conidial suspensions from each strain was placed on a hydrophobic cover glass. After incubating in darkness at 27 °C for 22 h, replace the sterile distilled water containing appressoria with a glycerol solution of concentration ranging from 1 to 4 M. Finally, after 10 min of incubation, the rate of appressorial collapse was observed and recorded under the microscope. The collapse rate of at least 100 appressoria per strain should be assessed.

### 4.6. Transmission Electron and Confocal Microscopy

To observe melanin dots on the cell wall. WT, *Chpks*Δ and *Chthr1*Δ strains were cultured in PDB liquid medium for a duration of 10 days prior to conducting transmission electron analysis. Subsequently, hyphae from WT, *Chpks*Δ and *Chthr1*Δ strains were collected after 10 d of incubation in PDB and fixed in 2.5% (*v/v*) glutaraldehyde in 0.1 M sodium phosphate buffer (pH 7.3) at 4 °C for 24 h. They were encapsulated in 3% (*w/v*) low melting point agarose and then processed in Spurr resin using a Lynx tissue processor on a 24 h schedule. Additional permeation under vacuum at 60 °C was performed before embedding the samples and polymerizing at 60 °C for 48 h, followed by ultrathin sectioning. The observations were made using a Hitachi HT7700 Exalens microscope, alloweing for detailed examination at high resolution levels.

To observe the expression of ChThr1 during the infection of *C. higginsianum.* The conidia suspension (1 × 10^5^·mL^−1^) of *Chthr1*Δ-C was dropped onto the detached Chinese flowering cabbage leaves. The infected leaves were then incubated at a temperature of 27 °C in complete darkness. Fluorescence was observed at 8 h post-infection (hpi), 22 hpi, and 40 hpi using epifluorescence microscopy with a Nikon Instruments A1 Confocal Laser Microscope equipped with a Plan Apochromat 60×/1.27 oil immersion objective. Image analysis was performed using Fiji 1.53q and Affinity Photo 2.0.3 software.

### 4.7. Pathogenicity Analysis

To assess the pathogenicity of the strains, conidia suspensions (1 × 10^6^·mL^−1^) were prepared and collected as described previously. These suspensions were then evenly sprayed onto *Arabidopsis* leaves and subjecting them to a controlled light/dark cycle lasting for 12 h each day at a temperature of 27 °C, *Arabidopsis* plants were observed and analyzed 4 d after inoculation. Additionally, under constant temperature conditions of 27 °C and in a completely dark environment, 20 μL of conidia suspension was drop onto wounded and unwounded detached Chinese flower cabbage leaves. The Chinese flowering cabbage leaves were observed and analyzed 4 d after inoculation.

## Figures and Tables

**Figure 1 ijms-24-15890-f001:**
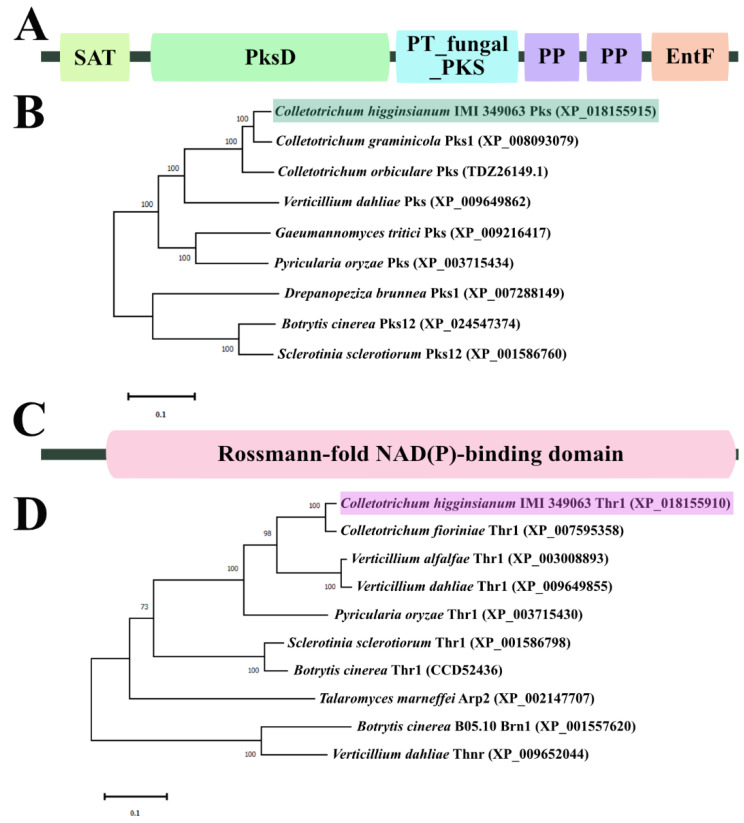
Bioinformatics analyses of ChPks and ChThr1 in *Colletotrichum higginsianum*. (**A**) Pfam domain analysis of ChPks, including SAT, PksD, PT_fungal_PKS, two PP domains, and EntF domain. (**B**) Phylogenetic tree analysis of ChPks and its homologues in *C. graminicola*, *C. orbiculare*, *V*. *dahliae*, *Gaeumannomyces tritici*, *Pyricularia oryzae*, *Drepanopeziza brunnea*, *B*. *cinerea* and *Sclerotinia sclerotiorum.* (**C**) Pfam domain analysis of ChThr1 with a Rossmann-fold NAD(P)-binding domain. (**D**) Phylogenetic tree analysis of ChThr1 and its homologues in *C. fioriniae*, *V. alfalfae*, *V. dahliae*, *P. oryzae*, *S. sclerotiorum*, *B. cinerea* and *Talaromyces marneffei*, along with the Thnr homologues in *B. cinerea* and *V. dahliae*.

**Figure 2 ijms-24-15890-f002:**
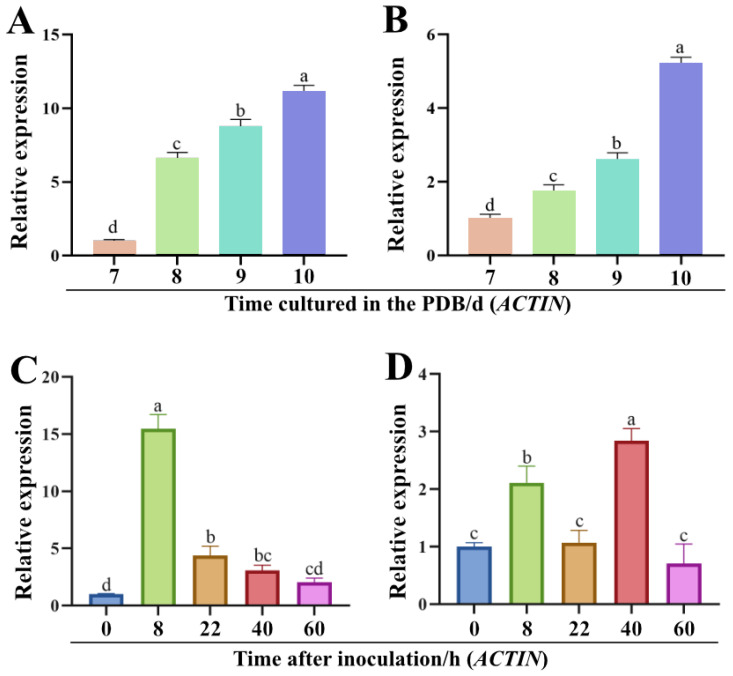
Expression analyses of genes *ChPKS* and *ChTHR1* during hypha and appressorium melanization. (**A**,**B**) Expression patterns of genes *ChPKS* and *ChTHR1* during hypha melanization with *ACTIN* as the endogenous reference gene. (**C**,**D**) Expression patterns of genes *ChPKS* and *ChTHR1* during appressorium melanization with *ACTIN* as the endogenous reference gene. Error bars represent standard deviations from three replicates, experimental data were analyzed by one-way ANOVA. Different letters indicate a significant difference at *p* < 0.05.

**Figure 3 ijms-24-15890-f003:**
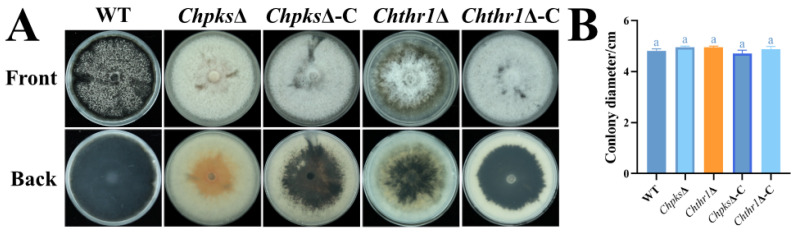
Defects in hyphae melanization of the *Chpks*Δ and *Chthr1*Δ mutants. (**A**) Colony morphology of the WT, *Chpks*Δ, *Chthr1*Δ, *Chpks*Δ-C and *Chthr1*Δ-C strains grown on PDA. (**B**) Colony diameters for the indicated strains on PDA. Error bars represent standard deviations from three replicates. The experimental data were analyzed using one-way ANOVA. Different letters indicate a significant difference at *p* < 0.05.

**Figure 4 ijms-24-15890-f004:**
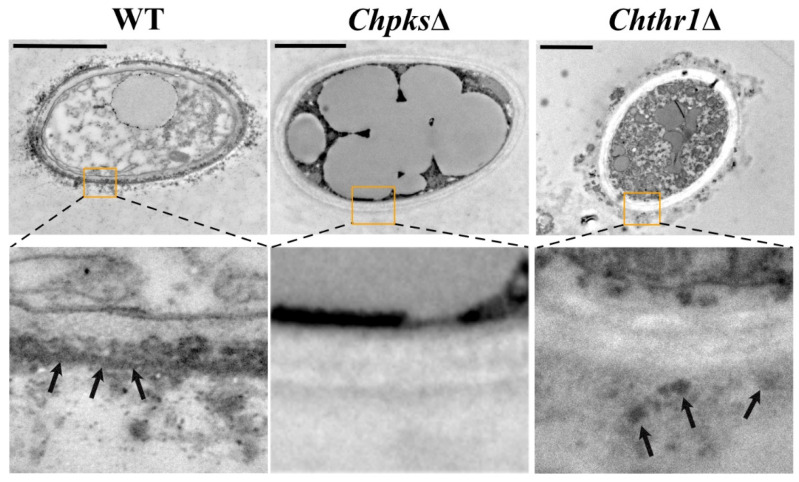
TEM (transmission electron microscopy) analysis of WT, *Chpks*Δ and *Chthr1*Δ strains. The images below each show the region enclosed by the enlarged orange box. Arrows indicate cell wall melanin deposition, and bars represent a scale of 2 μm.

**Figure 5 ijms-24-15890-f005:**
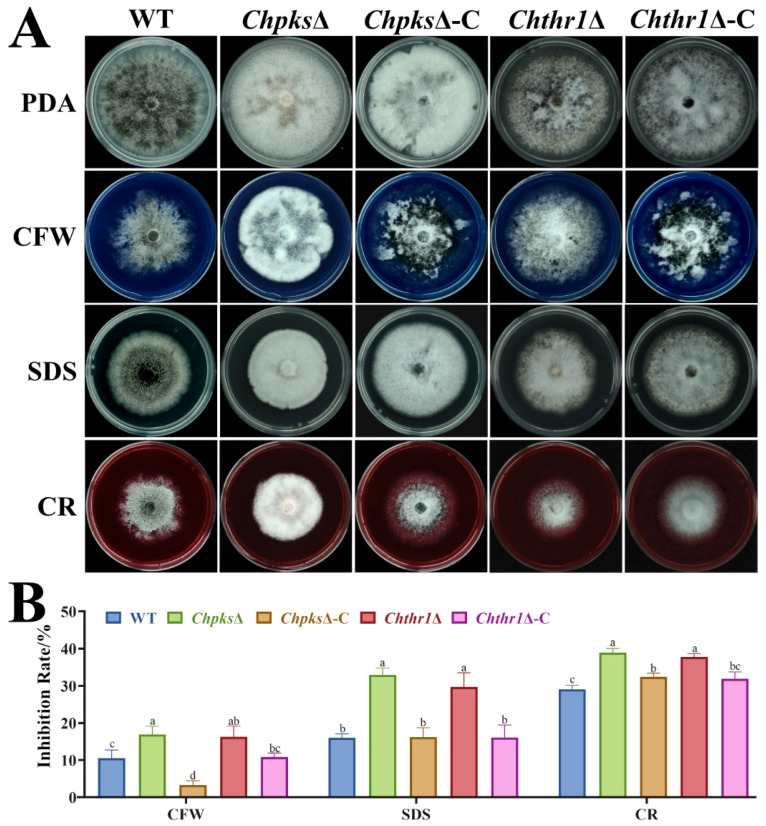
Sensitivity analysis of the *Chpks*Δ and *Chthr1*Δ mutants to cell wall stresses. (**A**) Colony morphology of the WT, *Chpks*Δ, *Chthr1*Δ, *Chpks*Δ-C and *Chthr1*Δ-C strains grown on PDA with 200 μg·mL^−1^ calcofluor white (CFW), 0.01% SDS, and 200 μg·mL^−1^ Congo red (CR). (**B**) Statistical analysis of inhibition rates of these strains under different stress conditions. Error bars represent standard deviations from three replicates. The experimental data were analyzed by one-way ANOVA. Different letters indicate significant difference at *p* < 0.05.

**Figure 6 ijms-24-15890-f006:**
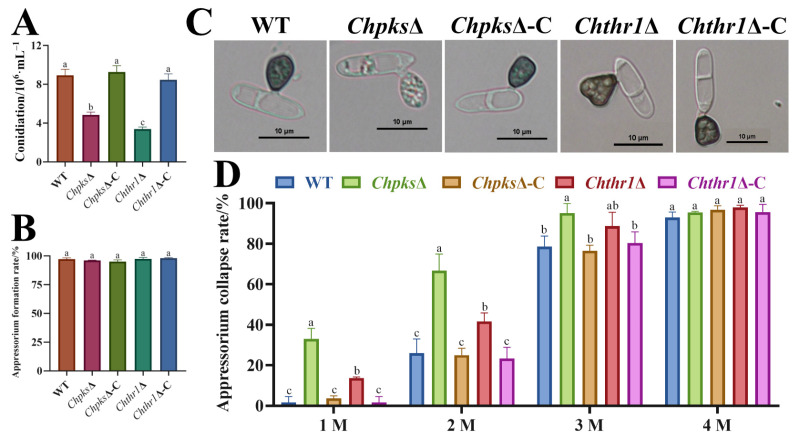
Conidiation, appressorium formation, morphology, and turgor pressure of the *Chpks*Δ and *Chthr1*Δ mutants. (**A**) Conidiation of the WT, *Chpks*Δ, *Chthr1*Δ, *Chpks*Δ-C and *Chthr1*Δ-C strains grown on mathur medium for 5 d. (**B**) Appressorium formation rates of the WT, *Chpks*Δ, *Chthr1*Δ, *Chpks*Δ-C and *Chthr1*Δ-C strains. (**C**) Appressorium morphology of the WT, *Chpks*Δ, *Chthr1*Δ, *Chpks*Δ-C and *Chthr1*Δ-C strains. (**D**) Appressorium collapse rates of the WT, *Chpks*Δ, *Chthr1*Δ, *Chpks*Δ-C and *Chthr1*Δ-C strains under glycerol concentration from 1 M to 4 M. Error bars represent standard deviations from three replicates. The experimental data were analyzed by one-way ANOVA. Different letters indicate significant difference at *p* < 0.05.

**Figure 7 ijms-24-15890-f007:**
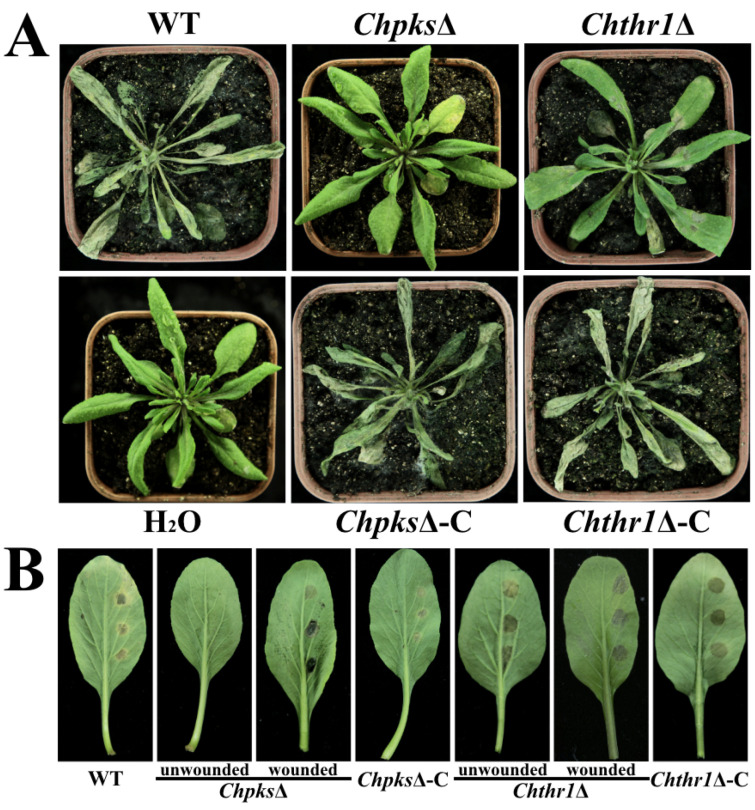
Analysis of pathogenicity of the *Chpks*Δ and *Chthr1*Δ mutants. (**A**) Symptoms observed on *Arabidopsis thaliana* plants 5 days post-inoculation with conidial suspensions of the WT, *Chpks*Δ, *Chthr1*Δ, *Chpks*Δ-C and *Chthr1*Δ-C strains. (**B**) Symptoms observed on detached leaves of Chinese flowering cabbage 4 days post-inoculation with conidial suspensions of the WT, *Chpks*Δ, *Chthr1*Δ, *Chpks*Δ-C and *Chthr1*Δ-C strains.

**Table 1 ijms-24-15890-t001:** List of primers used in this study.

Primer Name	Sequence (5′~3′)
ACTIN	CCCCAAGTCCAACAGAGAGA
	CATCAGGTAGTCGGTCAAGTCA
β-TUBULIN	GCCCTATTCTCGCTCGTCTTCC
	GGGCTCCAAATCGCAGTAAATG
PKS-qP	GTTCAAGTGGTTCTCATGGCTT
	CTGGCGATGGGGATGTAATTAG
THR1-qP	ACTTTGTCATCTCCAACTCGG
	ATGGAGGAGGTGAGGAGGAT
THNR-qP	GCACCATCGAGACATTTGT
	GGTACATGTCGGTCTTGATG
PKS-UP	ACGACGGCCAGTGCCAAGCTTCTGCTTTCAAACTCGTTTCACG
	GACCTGCAGGCATGCAAGCTCAAGAACAAGATTTCTGGATGTCG
PKS-DS	GACTCTAGAACTAGTGGATCCGGAACTTCATAGCGAAGCATATGA
	AGCTCGGTACCCGGGGGATCCGGAGTATTCGGGGAAGGAGAAT
THR1-UP	ACGACGGCCAGTGCCAAGCTTTTAAGCCAAAAAACGACTTGATAGA
	GACCTGCAGGCATGCAAGCTTAGTTGCTATGAAAAGACTGTGGCG
THR1-DS	CCGGGTACCGAGCTCGAATTCGTGCGATGATGCCACAAACTC
	TATGGAGAAACTCGAGAATTCGACACCGGGAGAGGGGAGG
PKSc-F1	GAAACTCGAGCTCGAGAATTCATCGGTACCCTATCTGGCTGC
	CCGGGTACCGAGCTCGAATTCGCTCGCTGGTTACCTCGTTCG
PKSc-F2	GACTCTAGATCTAGAGTCGACCTGCTTTCAAACTCGTTTCACG
	CTTGCATGCCTGCAGGTCGACTTACTCGCAAACCGCTTCACG
PKSc-F3	GACCTGCAGGCATGCAAGCTTGGAACTTCATAGCGAAGCATATGA
	ACGACGGCCAGTGCCAAGCTTGTAACGACAAAAAAAGCCACAGG
THR1c-F1	GAAACTCGAGCTCGAGAATTCCTCTAGCCGAAGGTATGAACCG
	CCGGGTACCGAGCTCGAATTCCCGCCGCCACGGAGGGGC
THR1c-F2	ACTCTAGATCTAGAGTCGACCGGTGGCGGCGGCATGGTCGCG
	TCACACCAGATCCGCCTGTGCCGCCACCGGTGACC
THR1c-F3	CACAGGCGGATCTGGTGTGAGCA
	GGCATCATCGCACTTACTTGTACAGCTCGTCCATGCC
THR1c-F4	CAAGTAAGTGCGATGATGCCACAAACTC
	CTTGCATGCCTGCAGGTCGACCCGACATCCGGGGAACAC
PKS	GCACCGGTACCCAGG
	GGTAGATGGTGTCGCTGG
THR1	ATGGCGCCCTCAGCGACTGAGAA
	CCTAGATATAAGCTATGTTAGGTATCCC
HPH1	GGGCGTCGGTTTCCACTAT
	GATATGTCCTGCGGGTAAATAGC
NeoR	GATAGAAGGCGATGCGCT
	ACCCGGTCATACCTTCTTAAG
PKS-probe	CACAGGACATTACATGACGCA
	CACCCTTGCTTATAACTTCGCA
THR1-probe	ACATAGCTTATATCTAGGACAGGCG
	TGTTACTAAACCTAGCTTGGGAAGG

## Data Availability

The data presented in this study are available on request from the corresponding author.

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
