# Peer review of "Deficiency of ChPks and ChThr1 Inhibited DHN-Melanin Biosynthesis, Disrupted Cell Wall Integrity and Attenuated Pathogenicity in Colletotrichum higginsianum"

_ijms, 2023, doi:10.3390/ijms242115890_

Round 1

Reviewer 1 Report

Comments and Suggestions for Authors

The manuscript “Deficiency of ChPks and ChThr1 Inhibited DHN-melanin Biosysthesis, Disrupted Cell Wall Integrity and Attenuated Pathogenicity in Colletotrichum higginsianum” presents an interesting study relating melanin biosynthesis with the pathogenicity of Ch to Brassica parachi. The study is very interesting and the authors presented it in a clear way. 

However, there are some issues that should be improved and complemented with some additional work listed below. 

The authors should:

1 – Support the results and discussion in microscopy evaluation of the infection process, only with this the authors could support the relation between ChTHR1 with biotrophy…

(Why do the authors consider that ChTHR1 is involved in biotrophy since by 40hai it is not clear if the infection strategy is still the biotrophy? The authors should complete the studies with a microscopy approach to clarify the relationship between the overexpression of ChTHR1 and the infection process.)

2 – Complement the expression studies with the analyses along the infection process with the mutants and the mutants complemented with the knock-out gene. It would be very interesting to analyze the expression profile of these 2 genes along the infection process of the ChPks or ChTHR1 as well as ChPks-C or ChTHR1-C.

3 – Use more than one reference gene as indicated by MIQE “The MIQE Guidelines: Minimum Information for Publication of Quantitative Real-Time PCR Experiment” (Bustin et al 2009) (https://doi.org/10.1373/clinchem.2008.112797)

4 – Explain properly the methodology namely deletion and complementation of ChPKS and ChTHR1; as well as the evaluation of conidiation and appressoria formation and the ability to turgor pressure

Minor issues

Figure 1 - Phylogenetic tree analysis of ChPks and ChThr1: Why the authors did not use the same group of fungus? Since it is assumed that both enzymes are present at the same pathway it would be interesting to see if the main phylogenetic relationship is conserved.

Figure 2 – The caption should be clarified since it is not clear what the authors did. It should be better explained that charts A and C show the expression profile along the mycelia growth and charts B and D show the expression profile along the infection process.

The colony diameters should be measured during colony growth, and the results presented should be taken before the colony reaches the edge of the plate.

Figure 5 – Inhibition  not inhabition

Inhiinhibinhibit

Author Response

Response to Reviewer 1 Comments

(Manuscript ID: ijms-2660389-R1)

Dear Editor and Reviewers,

Thank you for granting us the privilege to submit a revised version of our manuscript entitled “Deficiency of ChPks and ChThr1 Inhibited DHN-melanin Biosynthesis, Disrupted Cell Wall Integrity and Attenuated Pathogenicity in Colletotrichum higginsianum” to your esteemed journal “IJMS”. We are sincerely grateful for the time and effort invested in reviewing our manuscript and for the valuable feedback and insightful comments provided. We greatly appreciate your contributions, which have significantly enhanced the quality of our paper.

We have carefully incorporated the suggestions made by the reviewers and have made the necessary changes in the revised manuscript. These changes have been highlighted in red color for easy identification. Below, you will find a point-by-point response to the comments and concerns raised by the reviewers. The page numbers refer to the revised version of the manuscript with tracked changes (ijms-2660389-R1).

Comments 1: Support the results and discussion in microscopy evaluation of the infection process, only with this the authors could support the relation between ChTHR1 with biotrophy…

(Why do the authors consider that ChTHR1 is involved in biotrophy since by 40hai it is not clear if the infection strategy is still the biotrophy? The authors should complete the studies with a microscopy approach to clarify the relationship between the overexpression of ChTHR1 and the infection process.)

Response 1: Thank you for your thoughtful feedback. Previous research has demonstrated that Colletotrichum higginsianum can penetrate the epidermal cells of host plant around 40 hours after inoculation, indicating the onset of the biotrophic infection phase. In our study, we observed a significant upregulation of gene ChTHR1 expression at 40 hours post-inoculation, suggesting its potential involvement in the biotrophic infection phase. To further investigate this, we have included a fluorescence observation assay of ChThr1-mCherry in our revised manuscript. This assay allowed us to analyze the expression of ChTHR1 during the infection process of C. higginsianum and examine its relationship with the biotrophic infection phase. More details on this experiment can be found in subsections 2.2 and 4.6, as well as in Figure S2. In our future studies, we plan to delve deeper into the function of ChThr1 in C. higginsianum.

Comments 2: Complement the expression studies with the analyses along the infection process with the mutants and the mutants complemented with the knock-out gene. It would be very interesting to analyze the expression profile of these 2 genes along the infection process of the ChpksΔ or Chthr1Δ as well as ChpksΔ-C or Chthr1Δ-C.

Response 2: Thank you for your suggestion. We also find it very interesting to analyze the expression profile of these two genes during the infection process of the ChpksΔ or Chthr1Δ mutants, as well as ChpksΔ-C or Chthr1Δ-C mutants. However, conducting this experiment would require a longer time frame, and therefore we wouldn't be able to obtain the results within 10 days. In future experiments, we will make sure to address this question.

Comments 3: Use more than one reference gene as indicated by MIQE “The MIQE Guidelines: Minimum Information for Publication of Quantitative Real-Time PCR Experiment” (Bustin et al 2009) (https://doi.org/10.1373/clinchem.2008.112797).

Response 3: Thank you for the kind reminder. We have made the necessary revisions to the method of RT-qPCR according to the MIQE Guidelines. We included more details on the expression patterns of genes ChPKS and ChTHR1 in hypha and appressorium melanization, using β-TUBULIN as the endogenous reference gene. We also analyzed the expression of gene ChTHNR in WT and Chthr1Δ using β-TUBULIN as the endogenous reference gene. Please refer to subsection 2.2, 4.6, Figure S2, and Figure S5 for more information.

Comments 4: Explain properly the methodology namely deletion and complementation of ChPKS and ChTHR1; as well as the evaluation of conidiation and appressoria formation and the ability to turgor pressure.

Response 4: Thank you for your valuable comments. We have taken your comments into consideration and made the necessary revisions to our manuscript. In particular, we have added more details to the methods section regarding the deletion and complementation of genes ChPKS and ChTHR1, as well as the analyses of conidiation, appressorial formation, and appressorial turgor. Please refer to subsections 4.4 and 4.5 for the updated information.

Comments 5: Figure 1 - Phylogenetic tree analysis of ChPks and ChThr1: Why the authors did not use the same group of fungus? Since it is assumed that both enzymes are present at the same pathway it would be interesting to see if the main phylogenetic relationship is conserved.

Response 5: Thank you for your suggestion. We followed your advice and searched for relevant reports on Colletotrichum species that could be compared to C. higginsianum using BLAST. However, we were unable to find any reports specifically on THR1 in Colletotrichum. As a result, we decided to use THR1 from Sclerotinia sclerotiorum as a comparison in our study with C. higginsianum.

Comments 6: Figure 2 – The caption should be clarified since it is not clear what the authors did. It should be better explained that charts A and C show the expression profile along the mycelia growth and charts B and D show the expression profile along the infection process

Response 6: Thank you for pointing out the issue with Figures 2B and 2C in Figure 2 of our manuscript. We apologize for the oversight. We have now adjusted the positioning of Figures 2B and 2C accordingly. Additionally, we have included information about the sample treatment conditions and the endogenous reference gene in Figure 2. Please refer to the revised Figure 2 for the updated details.

Comments 7: The colony diameters should be measured during colony growth, and the results presented should be taken before the colony reaches the edge of the plate.

Response 7: Thank you for your thoughtful feedback. We appreciate your comment regarding the choice of time for determining the colony diameter of C. higginsianum. We had chosen 5 days at which C. higginsianum typically reaches full growth on a 6 cm PDA plate. We will definitely take note of this and consider it for our future experiments. Thank you again for your valuable input.

Comments 8: Figure 5 – Inhibition not inhabition

Response 8: Thank you for pointing out this error. We have made the necessary revisions to the manuscript, specifically in Figure 5 and subsection 3. Please review the updated version and let us know if you have any further questions or concerns.

Taken together, we have made every effort to revise and improve the manuscript based on the provided comments and suggestions. It is important to note that these modifications and changes have been implemented without altering the paper's content and structure.

We sincerely appreciate your diligent work and sincerely hope that the corrections we have made will meet with your approval.

Once again, thank you very much for your comments and suggestions.

Sincerely,

Authors: Lingtao Duan, Li Wang, Yiming Zhu and Erxun Zhou, et al.

Reviewer 2 Report

Comments and Suggestions for Authors

The article contains a large body of evidence, but it provides only some new insights into the infection process of Colletotrichum higginsianum. The expression of ChPKS and ChTHR1 genes encoding enzymes related to DHN-melanin biosynthesis in C. higginsianum was analyzed in the manuscript. The Authors used the RT-qPCR technique and did not perform a whole-transcriptome analysis with sequencing. The advantage of the study is that molecular analyses were accompanied by pathogenicity tests and a microscopic analysis of the pathogen’s infection structures. The article will have numerous readers among persons who investigate infection processes in biotrophic pathogens. The background information provided in the Introduction section is sufficient, but the description of research methods is simplified – they should be described in greater detail, referring to previously published studies that contain detailed descriptions of the applied analytical techniques and procedures. Therefore, the methodology section requires additional clarification.

Title: replace “Biosysthesis” with “Biosynthesis”

P. 3., L. 3: Figure 1B should be placed before Figure 1D.

Figure 2. It is unclear whether Figures 2A and C present the gene expression of fungi growing on media, and Figures 2B and D present the gene expression of the pathogen after plant inoculation? The missing information should be provided in the description of figures.

Comments on subsection 4.1. Strains, Plant and Culture Conditions

Colletotrichum higginsianum strain IMI 349063 was described for the first time by O'Connell RJ et al., "Lifestyle transitions in plant pathogenic Colletotrichum fungi deciphered by genome and transcriptome analyses.", Nat Genet, 2012 Sep;44(9):1060-5

Did the Authors analyze one strain or several strains, as suggested in the sentence: “All fungal strains were cultured at a temperature of 27°C.”? Was one wild strain and several modifications of this strain analyzed?

How many biological replicates were used in the experiment?

Subsections 4.6 and 4.7: fragments of the research protocol, e.g. “After 4 days, observe and analyze”, should be modified.

Author Response

Response to Reviewer 2 Comments

(Manuscript ID: ijms-2660389-R1)

Dear Editor and Reviewers,

Thank you for granting us the privilege to submit a revised version of our manuscript entitled “Deficiency of ChPks and ChThr1 Inhibited DHN-melanin Biosynthesis, Disrupted Cell Wall Integrity and Attenuated Pathogenicity in Colletotrichum higginsianum” to your esteemed journal “IJMS”. We are sincerely grateful for the time and effort invested in reviewing our manuscript and for the valuable feedback and insightful comments provided. We greatly appreciate your contributions, which have significantly enhanced the quality of our paper.

We have carefully incorporated the suggestions made by the reviewers and have made the necessary changes in the revised manuscript. These changes have been highlighted in red color for easy identification. Below, you will find a point-by-point response to the comments and concerns raised by the reviewers. The page numbers refer to the revised version of the manuscript with tracked changes (ijms-2660389-R1).

Comments 1: The article contains a large body of evidence, but it provides only some new insights into the infection process of Colletotrichum higginsianum. The expression of ChPKS and ChTHR1 genes encoding enzymes related to DHN-melanin biosynthesis in C. higginsianum was analyzed in the manuscript. The Authors used the RT-qPCR technique and did not perform a whole-transcriptome analysis with sequencing. The advantage of the study is that molecular analyses were accompanied by pathogenicity tests and a microscopic analysis of the pathogen’s infection structures. The article will have numerous readers among persons who investigate infection processes in biotrophic pathogens. The background information provided in the Introduction section is sufficient, but the description of research methods is simplified – they should be described in greater detail, referring to previously published studies that contain detailed descriptions of the applied analytical techniques and procedures. Therefore, the methodology section requires additional clarification.

Response 1: Thank you for your thoughtful feedback. We have taken your comments into consideration and have made revisions to our applied analytical techniques and procedures. Specifically, we have provided a more detailed description in subsections 4.3, 4.4, 4.5, 4.6, and 4.7 of our manuscript. We appreciate your attention to detail and the opportunity to improve our work.

Comments 2: Title: replace “Biosysthesis” with “Biosynthesis”.

Response 2: Thank you for pointing out this error. We appreciate your feedback and have revised the title of our manuscript in the revised version.

Comments 3: P. 3., L. 3: Figure 1B should be placed before Figure 1D.

Response 3: Thank you for your suggestion. We have made the requested modification to Figure 1 by repositioning Figure 1B and Figure 1C. Now, Figure 1B appears before Figure 1D. Please take a look at the updated Figure 1.

Comments 4: Figure 2. It is unclear whether Figures 2A and C present the gene expression of fungi growing on media, and Figures 2B and D present the gene expression of the pathogen after plant inoculation? The missing information should be provided in the description of figures.

Response 4: Thank you for your valuable suggestion. We apologize for any confusion caused by the previous placement of Figures 2B and 2C in Figure 2. We have made the necessary adjustments as per your suggestion. Additionally, we have now included information regarding the sample treatment conditions and the endogenous reference gene in Figure 2. Please refer to the revised Figure 2 for these changes.

Comments 5: Comments on subsection 4.1. Strains, Plant and Culture Conditions

Colletotrichum higginsianum strain IMI 349063 was described for the first time by O'Connell RJ et al., "Lifestyle transitions in plant pathogenic Colletotrichum fungi deciphered by genome and transcriptome analyses.", Nat Genet, 2012 Sep;44(9):1060-5

Did the Authors analyze one strain or several strains, as suggested in the sentence: “All fungal strains were cultured at a temperature of 27°C.”? Was one wild strain and several modifications of this strain analyzed?

Response 5: Thank you for your valuable feedback. In this study, we conducted our research using a single strain, namely Colletotrichum higginsianum strain IMI 349063. All mutants used in this study were derived from C. higginsianum strain IMI 349063. Furthermore, the wild-type, mutant, and complemented strains were all cultured at a temperature of 27℃.

Comments 6: How many biological replicates were used in the experiment?

Response 6: Thank you for your careful comments. In this study, we have used three biological replicates in each experiment.

Comments 7: Subsections 4.6 and 4.7: fragments of the research protocol, e.g. “After 4 days, observe and analyze”, should be modified.

Response 7: Thank you for your suggestion. We have made revisions to certain sections of the research protocol, specifically subsections 4.6 and 4.7, in the updated manuscript. Please refer to subsections 4.6 and 4.7 for the modifications made.

Taken together, we have made every effort to revise and enhance the manuscript based on the provided comments and suggestions. It is important to note that these modifications and changes have been implemented without altering the paper's content and structure.

We sincerely appreciate your diligent work and sincerely hope that the corrections we have made will meet with your approval.

Once again, thank you very much for your comments and suggestions.

Sincerely,

Authors: Lingtao Duan, Li Wang, Yiming Zhu and Erxun Zhou, et al.

Reviewer 3 Report

Comments and Suggestions for Authors

The Manuscript entitled “Deficiency of ChPks and ChThr1 Inhibited DHN-melanin Biosysthesis, Disrupted Cell Wall Integrity and Attenuated Pathogenicity in Colletotrichum higginsianum” by Duan et al. explores the molecular mechanism underlying melanin biosynthesis in the fungus responsible for anthracnose in Chinese flowering cabbage. The authors focused their research on two enzymes (ChPks and ChThr1) related to the biosynthetic pathway by studying gene expression, melanin accumulation, pathogenicity and cell wall integrity in knockout mutants. In my opinion, the topic of interest and the manuscript is mostly well written, scientifically valid, and technically correct. The introduction covers the basic information related to the topic and the aim of the study. The presentation of results and discussion is clear. Materials and methods should be revised to improve clarity and details provided. There are several typos in the manuscript that should be corrected. I support the manuscript for publication in International Journal of Molecular Sciences after some revision. Few comments or suggested changes have been included in the attached revised manuscript.

Comments on the Quality of English Language

There are several typos and some unclear sentences in the manuscript that should be changed.

Author Response

Response to Reviewer 3 Comments

(Manuscript ID: ijms-2660389-R1)

Dear Editor and Reviewers,

Thank you for granting us the privilege to submit a revised version of our manuscript entitled “Deficiency of ChPks and ChThr1 Inhibited DHN-melanin Biosynthesis, Disrupted Cell Wall Integrity and Attenuated Pathogenicity in Colletotrichum higginsianum” to your esteemed journal “IJMS”. We are sincerely grateful for the time and effort invested in reviewing our manuscript and for the valuable feedback and insightful comments provided. We greatly appreciate your contributions, which have significantly enhanced the quality of our paper.

We have carefully incorporated the suggestions made by the reviewers and have made the necessary changes in the revised manuscript. These changes have been highlighted in red color for easy identification. Below, you will find a point-by-point response to the comments and concerns raised by the reviewers. The page numbers refer to the revised version of the manuscript with tracked changes (ijms-2660389-R1).

Comments: The Manuscript entitled “Deficiency of ChPks and ChThr1 Inhibited DHN-melanin Biosysthesis, Disrupted Cell Wall Integrity and Attenuated Pathogenicity in Colletotrichum higginsianum” by Duan et al. explores the molecular mechanism underlying melanin biosynthesis in the fungus responsible for anthracnose in Chinese flowering cabbage. The authors focused their research on two enzymes (ChPks and ChThr1) related to the biosynthetic pathway by studying gene expression, melanin accumulation, pathogenicity and cell wall integrity in knockout mutants. In my opinion, the topic of interest and the manuscript is mostly well written, scientifically valid, and technically correct. The introduction covers the basic information related to the topic and the aim of the study. The presentation of results and discussion is clear. Materials and methods should be revised to improve clarity and details provided. There are several typos in the manuscript that should be corrected. I support the manuscript for publication in International Journal of Molecular Sciences after some revision. Few comments or suggested changes have been included in the attached revised manuscript.

There are several typos and some unclear sentences in the manuscript that should be changed.

Thank you for your meticulous feedback. We have incorporated additional details into the materials and methods section of the revised manuscript. Furthermore, we have made adjustments to several words and sentences that were unclear in the revised manuscript.

Taken together, we have made every effort to revise and enhance the manuscript based on the provided comments and suggestions. It is important to note that these modifications and changes have been implemented without altering the paper's content and structure.

We sincerely appreciate your diligent work and sincerely hope that the corrections we have made will meet with your approval.

Once again, thank you very much for your comments and suggestions.

Sincerely,

Authors: Lingtao Duan, Li Wang, Yiming Zhu and Erxun Zhou, et al.